# Small RNA and mRNA Sequencing Reveal the Roles of microRNAs Involved in Pomegranate Female Sterility

**DOI:** 10.3390/ijms21020558

**Published:** 2020-01-15

**Authors:** Lina Chen, Xiang Luo, Xuanwen Yang, Dan Jing, Xiaocong Xia, Haoxian Li, Krishna Poudel, Shangyin Cao

**Affiliations:** Zhengzhou Fruit Research Institute, Chinese Academy of Agricultural Sciences, 450009 Zhengzhou, China; 82101171109@caas.cn (L.C.); luoxiang@caas.cn (X.L.); or 82101199219@caas.cn (X.Y.); 82101176054@caas.cn (D.J.); xiaxiaocong12@163.com (X.X.); lihaoxian@caas.cn (H.L.); or 2018Y090100100@caas.cn (K.P.)

**Keywords:** high-throughput sequencing, functional male flowers, bisexual flowers, reproduction process, andromonoecy, ovule

## Abstract

Female sterility is a key factor restricting plant reproduction. Our previous studies have revealed that pomegranate female sterility mainly arose from the abnormality of ovule development. MicroRNAs (miRNAs) play important roles in ovule development. However, little is known about the roles of miRNAs in female sterility. In this study, a combined high-throughput sequencing approach was used to investigate the miRNAs and their targeted transcripts involved in female development. A total of 103 conserved and 58 novel miRNAs were identified. Comparative profiling indicated that the expression of 43 known miRNAs and 14 novel miRNAs were differentially expressed between functional male flowers (FMFs) and bisexual flowers (BFs), 30 known miRNAs and nine novel miRNAs showed significant differences among different stages of BFs, and 20 known miRNAs and 18 novel miRNAs exhibited remarkable expression differences among different stages of FMFs. Gene ontology (GO) analyses of 144 predicted targets of differentially expressed miRNAs indicated that the “reproduction process” and “floral whorl development” processes were significantly enriched. The miRNA–mRNA interaction analyses revealed six pairs of candidate miRNAs and their targets associated with female sterility. Interestingly, pg-miR166a-3p was accumulated, whereas its predicted targets (*Gglean012177.1* and *Gglean013966.1*) were repressed in functional male flowers (FMFs), and the interaction between pg-miR166a-3p and its targets (*Gglean012177.1* and *Gglean013966.1*) were confirmed by transient assay. *A. thaliana* transformed with 35S-pre-pg-miR166a-3p verified the role of pg-miR166a-3p in ovule development, which indicated pg-miR166a-3p’s potential role in pomegranate female sterility. The results provide new insights into molecular mechanisms underlying the female sterility at the miRNA level.

## 1. Introduction

Female sterility is a widespread phenomenon in flowering plants restricting plant reproduction [1,2,3]. Abortion of either sexual organs is the initial step towards a sexually dimorphic specie [4]. Pomegranate (*Punica granatum* L.), one of the important deciduous fruit crops worldwide, is valued for its significant medicinal and economic values [5,6]. Pomegranate is characterized by andromonoecy, i.e., two types of flowers are produced on the same plant, including the functional male flowers (FMFs) and bisexual flowers (BFs) [7,8]. The FMFs, which are referred as “infertile” and “bell” flowers, have well-developed male parts but abnormal female parts and they fail to set fruits and eventually fall from trees, whereas BFs, which are referred to as “fertile” and “vase-shaped” flowers, have well-formed female and male parts and can set fruits [7,8]. Moreover, pomegranate flowers can appear to be solitary, paired, or clustered [7]. The solitary flowers almost appear on spurs along the branches, while the clusters are terminal [7]. As previously reported, the ratio of BFs was distinct among different types of flowers [9]. The ratio of BFs in clusters was noticeably less than that in solitary and paired ones [9] (Figure 1a). Clustered terminal flowers contained more BFs than lateral flowers [9]. Although the proportion of FMFs is negatively correlated with crop productivity and yield, pomegranate female sterility can be helpful to optimize the allocation of limited nutritional resources to male and female functions in the evolutionary process [8]. In pomegranate, ovules are anatropous and bi-integument. The FMFs’ female sterility is closely related to abnormal ovule development, and their ovule development mainly ceases following the formation of the inner integument primordium [10].

MicroRNAs (miRNAs) are endogenous 20 to 24 nt, non-coding RNAs that are derived from primary miRNA transcripts (pri-miRNAs) containing a stem-loop secondary structure [11,12], functioning by suppressing the expression of specifically targeted genes at a post-transcriptional level through mRNA cleavage or translational inhibition [11,13]. In plants, miRNAs play crucial roles in plant organ development [14], stress tolerance [15], phytohormone signaling [16], growth phase change [17,18], and disease resistance [15]. The ovule development undergoes four main stages which include: initiation of primordia; regionalization of primordia into three regions, i.e., funiculus, chalaza, and nucellus; integuments development; and embryo sac development [19,20,21]. Many miRNAs have been shown to be important regulatory factors in ovule development [12,18]. For example, miR167 is essential for correct patterning of gene expression during ovule development in *A. thaliana*, which regulates ovule growth by limiting *Auxin Response Factors 6* (*ARF6*) and *Auxin Response Factors 8* (*ARF8*) transcript expression domains in cells that develop into integuments [22]. In addition, miR166/165-insensitive *phb-1d/+* mutant show arrested outer integument in *A. thaliana* [23] and MiR165/6 regulates ovule development by restricting the expression of *PHABULOSA* (*PHB*), is a member of class III homeodomain-leucine zipper (HD-ZIP III) and has important roles in inner integument development [24,25]. Moreover, miR156 and miR157 regulate ovule development by targeting *SQUAMOSA-promoter binding protein* (*SPL/SBP*) box transcription factors [26,27].

The study of small RNAs in pomegranate has been reported previously [28,29,30]. However, the involvement of small RNAs in pomegranate FMFs’ ovule development cessation has not yet been determined. The key stage in the termination of FMFs’ ovule development was when its bud vertical diameter was 5.1 to 13.0 mm [10]. Genes influencing ovule development such as *INNER NO OUTER* (*INO*) and *AINTEGUMENTA* (*ANT*) were identified as candidate genes affecting pomegranate female sterility by RNA-seq [10]. To investigate miRNA-target modules involved in female sterility in pomegranate, sRNA-seq was conducted in pistils between three pairs of FMFs and BFs’ pistils prior to the appearance of, and subsequent to, ovule inner integument primordium formation. Finally, the candidate miRNA-target modules that influence pomegranate female sterility were identified, which contributed to enhancing the understanding of pomegranate ovule development.

## 2. Results

### 2.1. Deep Sequencing of Small RNAs of Pomegranate Fertile and Sterile Pistils

A total of 18 small RNA sequencing data were obtained which were deposited in the figshare database(https://figshare.com/articles/Small_RNA_sequencing_identify_miRNAs_involved_in_pomegranate_female_sterility/9563579). Through sequencing of 18 samples, a total of 409.8 Gb clean reads were generated, and the number of reads yielded from the 18 small RNA sequencing libraries ranged from 22.3 to 23.1 million. After aligning against the pomegranate reference genome, 84.8% to 89.6% of the reads in 18 libraries were mapped to the pomegranate reference genome [31] (Appendix A), with CG content that varied from 47.1% to 50.1% (Appendix A). The number of unique sRNAs for 18 accessions ranged from 8.4 to 10.2 million, and 77% to 82.2% were mapped to the pomegranate reference genome [31] (Appendix A). Small RNAs of 20 to 24 nucleotides (nt) in length were dominant in all 18 sequencing libraries (Figure 1b), of which the 24 nt length small RNAs showed a large percentage in the small RNA libraries (Figure 1b) and the 21 nt length small RNAs also showed a large percentage in the small RNA libraries (Figure 1b), suggesting the existence of post-transcription during pomegranate ovule development. 

### 2.2. Identifying miRNAs Involved in Pomegranate Female Sterility

A final set of 103 known miRNAs were identified from pomegranate BFs and FMFs’ pistils, including miR858, miR166/165, miR160, miR157, miR159, and miR408 families, etc. (Appendix A). In addition, 58 novel miRNAs in pomegranate BFs and FMFs were also identified (Appendix A). Among all identified miRNAs, 43 known miRNAs and 14 novel miRNAs were differentially expressed in a comparison of FMFs and BFs (ATNSI_TNSI, ATNSII_TNSII, ATNSIII_TNSIII) (Appendix A and Figure 1c), indicating the miRNAs that were related to pomegranate female sterility. A total of 30 of the identified known miRNAs and nine of the novel miRNAs showed significantly different expression among different stages of BFs (TNSII_TNSI, TNSIII_TNSI, TNSIII_TNSII) (Appendix A and Figure 1d). Moreover, a total of 20 known miRNAs and 18 novel miRNAs exhibited remarkable expression differences among different stages of FMFs (ATNSII_ATNSI, ATNSIII_ATNSI, ATNSIII_ATNSII) (Appendix A and Figure 1e).

During different development stages of FMFs, 18 of the differentially expressed miRNAs showed significantly higher expression level in ATNSI as compared with ATNSII and ATNSIII (Figure 2a), including two ATNSI specific miRNAs, pg-miRN30 and pg-miRN36 (Figure 2a). Seven of the differentially expressed miRNAs showed significantly higher expression level in ATNSII as compared with ATNSI and ATNSIII, including pg-miR444b.1 and pg-miR528-5p that were only identified in the ATNSII (Figure 2a). The other 13 differentially expressed miRNAs were upregulated in ATNSIII as compared with ATNSI and ATNSII, in which pg-miR6105b was specifically expressed in ATNSIII and pg-miR166a-3p was abundantly expressed with read counts greater than 100,000 during FMFs’ pistil development (Appendix A and Figure 2a).

During different development stages of BFs, 15 differentially expressed miRNAs were upregulated in TNSI as compared with TNSII and TNSIII, of which pg-miRN18 and pg-miR7533a were specifically expressed in TNSI (Appendix A and Figure 2b). Moreover, seven differentially expressed miRNAs showed higher expression level in TNSII than that of TNSI and TNSIII (Figure 2b). The other 17 differentially expressed miRNAs manifested upregulation in TNSIII (Figure 2b).

Among the 57 differentially expressed miRNAs between BFs and FMFs’ pistils, 15 showed differential expression as compared with TNSI and ATNSI (Figure 2c), in which pg-miR5671a, pg-miR7533a, pg-miRN56, pg-miR528-5p, and pg-miR7717a-5p showed specific expression in TNSI (Appendix A and Figure 2c). In the comparison of TNSII and ATNSII, 10 miRNAs, including pg-miR397a and pg-miR408-3p which were abundantly expressed, were upregulated in ATNSII (Appendix A and Figure 2c), on the contrary, eight miRNAs, including pg-miR160a-3p, pg-miR858b, and pg-miRN07 which were highly expressed, exhibited significantly down-regulated in ATNSII as compared with TNSII (Appendix A and Figure 2c). Of the 27 differentially expressed miRNAs as compared with ATNSIII and TNSIII, 11 miRNAs were upregulated in ATNSIII, of which pg-miR398b, pg-miR166a-3p, and pg-miR902j-5p were highly expressed (Appendix A and Figure 2c). The other 16 miRNAs were downregulated in ATNSIII (Appendix A and Figure 2c).

### 2.3. Validation of Expression Patterns of miRNAs

To further confirm the miRNA sequencing results, the expression of 10 randomly selected miRNAs (pg-miR1514a-3p, pg-miR5671a, pg-miR4414 b, pg-miR408-3p, pg-miR398b, pg-miR398a-3p, pg-miR166a-3p, pg-miR397a, pg-miR167f-3p, and pg-miR160a-3p) were validated using stem-loop qRT-PCR. The results showed that the consistency rate was 80%. The expression patterns of miRNAs were largely consistent with those determined by high-throughput sequencing (Figure 3), implying the reliability of miRNA sequencing data. 

### 2.4. Identification and Analysis of miRNA Targets

A total of 208 and 177 target genes were identified for the conserved miRNAs and novel miRNAs, respectively (Appendix A), of which 144 were targeted by the differentially expressed miRNAs. Some novel miRNAs targeted the same gene families as the conserved miRNAs (Appendix A). For example, both pg-miRN07 and pg-miR166a-3p targeted *HD-ZIP* genes, *Gglean013966.1* and *Gglean012177.1* and pg-miRN01 and pg-miR858b targeted *MYB* family genes (Appendix A). 

To better understand the roles of miRNA in pomegranate female sterility, 144 targets of the differentially expressed miRNAs were used for the GO analysis. A total of 19 biological processes, 11 molecular functions, and two cellular components were significantly enriched (correction value at *p* < 0.01) (Appendix A). According to the biological processes, the overrepresented terms were reproduction process (GO:0000003, GO:0022414, and GO:0003006), post-embryonic development (GO:0009791), lignin catabolic process (GO:0046274, GO:0046271, and GO:0009808, GO:0009698), and floral whorl development (GO:0048438 and GO:0048437) (Appendix A). According to the molecular function, cation binding (GO:0043169), ion binding (GO:0046872, GO:0043167), copper ion binding (GO:0005507), and transition metal ion binding (GO:0046914) were prominently enriched (Appendix A). As shown in Appendix A, under the category of cellular component, apoplast (GO:0048046, GO:0005576) was significantly enriched.

### 2.5. MicroRNA–mRNA Interaction Identification

To identify the potential miRNA–mRNA regulatory network that could be related to pomegranate female sterility, the expression patterns of 144 genes targeted by the differentially expressed miRNAs were retrieved from the RNA-seq data (accession numbers SRX2735567-SRX2735584) (Appendix A). The miRNAs are known to suppress the expression of target genes by mRNA cleavage or translational inhibition [32]. To explore the roles of miRNA and mRNA interactions in female sterility, the negative correlation in miRNA-target modules was studied (Table 1). The pg-miR858b and pg-miRN01 co-repressed the expression level of *Gglean012452.1* in ATNSI, and pg-miR858b and pg-miRN01 showed upregulated in ATNSI between ATNSI and TNSI, whereas *Gglean012452.1* was downregulated in ATNSI (Table 1). Between ATNSII and TNSII, pg-miRN11 and pg-miR858b exhibited higher seq-freqs in TNSII and their targets (*Gglean007369.1*, *Gglean014673.1*, *Gglean008242.1*, *Gglean018250.1*, *Gglean023939.1*, and *Gglean004315.1*) showed opposite expression levels (Table 1). In addition, pg-miR165a-3p negatively regulated the expression of *Gglean013966.1* and *Gglean012177.1*, and pg-miR444b.1 repressed the expression of *Gglean003233.1* during the key stage of pomegranate female sterility (ATNSII_TNSII) (Table 1). Three negatively correlated miRNA-target modules were found between ATNSIII and TNSIII (Table 1), including pg-miR166a-3p-*Gglean012177.1*, pg-miR166a-3p-*Gglean013966.1*, and pg-miR952b-*Gglean018134.1*. Pg-miR858b, pg-miRN11, pg-miR165/166a-3p, pg-miR444b.1, pg-miR952b, and their targets were mined as candidate factors influencing pomegranate female development. These results highlighted the complicated interactions between miRNAs and mRNAs during pistil development in pomegranate.

### 2.6. Pg-miR166a-3p Functions via Target HD-Zip Transcription Factor

As mentioned above, pg-miR166a-3p was highly active during the pistil development of BFs and FMFs, and it was accumulated at a higher level in the pistil of FMFs than that of BFs (Figure 4a). *A. thaliana* plants transformed with pg-miR166a-3p driven by the CaMV35S promoter exhibited declining plant biomass, clip size, and seeds number (Figure 4b). The expression levels of pg-miR166a-3p in different lines were detected (Figure 4c). L2 (with relative high expression level) and L3 (with relative low expression level) were selected to observe the development of buds and ovules and count the number of the seeds. The results showed that the number of meristems of inflorescence increased in transformants (Figure 4d), whereas the number of ovule primordia decreased significantly (*p* < 0.01) (Figure 4d). The number of seeds was negatively correlated with the expression level of pg-miR166a-3p (Figure 4e–f). Taken together, the phenotypes observed in transgenic *A. thaliana* implied the role of pg-miR166a-3p in flower primordium and ovule development. The expressions of homology genes of *Gglean012177.1* (*PHB*) and *Gglean013966.1* (*PHV*) in *A. thaliana* were detected by qRT-PCR and the results showed that the expression of both *PHB* and *PHV* in L2 and L3 was significantly lower than in wild type, which indicated that both *PHB* and *PHV* were the putative targets of pg-miR166a-3p (Figure 4g).

In addition, pg-miR166a-3p and its predicted targeted genes *Gglean012177.1* and *Gglean013966.1* which were involved in reproductive structure development process (GO:0048608) showed reciprocal expression patterns during BFs and FMFs’ pistil development (Figure 4a). The expression levels of *Gglean012177.1* and *Gglean013966.1*, encoding the homeobox-leucine zipper (HD-ZIP) protein, gradually decreased from ATNSI to ATNSIII (Figure 4a), but HD-ZIP protein positively regulated ovule patterning and growth [33]. Collectively, the decreasing of HD-ZIP family genes (*Gglean012177.1* and *Gglean013966.1*) and increasing of pg-miR166a-3p expression level from ATNSI to ATNSIII could be related to the abnormal development of FMFs’ ovule. To verify the interactions between pg-miR166a-3p and two potential targets, *Gglean012177.1* and *Gglean013966.1*, in vivo, a transient assay was conducted in tobacco leaves (Figure 5). The interactions were determined by the significant reduction of GFP fluorescent signal intensity that resulted from co-expression of pg-miR166a-3p with their target sites in *Gglean012177.1* and *Gglean013966.1*. The results showed that both *Gglean012177.1* and *Gglean013966.1* could mediate cleavage by pg-miR166a-3p (Figure 5). A modified target site (inactive site) and a perfect complementary site were used as negative and positive control, respectively (Figure 5a,c). These results indicated that pg-miR166a-3p can interact with its targets (*Gglean012177.1* and *Gglean013966.1*) to affect the ovule development (Figure 5b,d).

### 2.7. Newly Identified miRNAs in Pomegranate

Several conserved and novel miRNAs in pomegranate have been identified in previous studies through high-throughput small RNA sequencing [28,29,30]. Compared with the miRNAs expressed in fruit [30], a total of 28 known miRNAs identified in this study were conserved in pistil and fruit, while 67 known miRNAs were newly identified in pistil. Of the 28 conserved miRNAs, pg-miR160a-5p, pg-miR166a-3p, pg-miR167h, pg-miR168a, pg-miR159a, pg-miR171b, and pg-miR319a-3p were abundantly enriched in both pistil and fruit, which indicated their pleiotropy in pistil and fruit development. The newly identified miRNAs, including one upregulated (pg-miR8712) and seven downregulated (pg-miR1514a-3p, pg-miR2592s-3p, pg-miR5671a, pg-miR6105b, pg-miR7533a, pg-miR7717a-5p, and pg-miR7782-3p) miRNAs as compared with ATNSI and TNSI; four upregulated (pg-miR444b.1, pg-miR528-5p, pg-miR6300, and pg-miR5671a) and three downregulated (pg-miR4414b, pg-miR6161a, and pg-miR8723a) miRNAs as compared with ATNSII and TNSII; and seven upregulated (pg-miR1077-5p, pg-miR393h, pg-miR403c-5p, pg-miR4376, pg-miR5152-3p, pg-miR902j-5p, and pg-miR952b) and six downregulated (pg-miR1172.1, pg-miR170-5p, pg-miR4240, pg-miR5223, pg-miR842-5p, and pg-miR8700) miRNAs as compared with ATNSIII and TNSIII. Those miRNAs including pg-miR166a-3p, pg-miR8712, pg-miR444b.1, pg-miR528-5p, pg-miR6300, pg-miR5671a, pg-miR1077-5p, pg-miR393h, pg-miR403c-5p, pg-miR4376, pg-miR5152-3p, pg-miR902j-5p, and pg-miR952b, upregulated in FMFs, could play important roles in pomegranate female sterility. 

## 3. Discussion

Female sterility is one of the key factors restricting pomegranate yield [8]. A previous study has explored many transcription factors that can affect pomegranate pistil development, of which many were miRNAs targets [9]. In order to improve our understanding of pomegranate female sterility, miRNAs expression profiles of developing FMFs and BFs’ pistil in pomegranate were performed, which was the first comprehensive analysis of miRNA expression pattern during pistil development in pomegranate. On the basis of the analyses of the differentially expressed miRNAs in pistil between FMFs and BFs at different stages, a strong association between pg-miR858b and pg-miRN01 and pistil development was identified at the early stage. The pg-miR444b.1, pg-miRN11, pg-miR166a/165a-3p, and pg-miR952b can influence pomegranate integument development at later stages. Specifically, overexpression of pg-miR166a-3p in *A. thaliana* exhibited declining plant biomass, clip length, and ovule number, implying the potential roles of pg-miR166a in pomegranate ovule development. 

### 3.1. Potential Roles of miRNAs in Pomegranate Female Sterility

BFs of pomegranate have fertile stamens and carpel, whereas FMFs have fertile stamens but arrested pistils (Figure 6). FMFs’ pistils mainly arrested after ovule inner tegument primordium formed [8,9]. Recently, increasing attention has been paid to the roles of miRNAs in ovule development, including *A. thaliana* [22], cotton [18], and rice [34]. In this study, a total of 161 miRNAs (including conserved and novel miRNAs) were identified in BFs and FMFs’ pistil.

It has been reported that miR858 is highly expressed in cotton ovules at zero and three days post anthesis, whereas its target gene *GhMYB2* was poorly expressed [35], which indicated the roles of miR858 and *GhMYB2* in cotton ovules development. Tomato *myb21* Cas9 mutants showed female sterility, which indicated a function of *MYB21* in tomato ovule development [36]. In this study, conserved pg-miR858b showed upregulation at early stages in FMFs, and its predicted target *Gglean012452.1*, which was annotated as the *MYB2* homology gene and exhibited the opposite pattern of expression (Figure 6 and Table 1), suggesting pg-miR858b and *Gglean012452.1* can have an impact on pomegranate female sterility. Moreover, the novel miRN01 also showed upregulation in ATNSI, and it targeted the same gene as miR858b (Figure 6 and Table 1). Therefore, miR858b and miRN01 may co-repress the expression of *Gglean012452.1* to affect the development of pomegranate ovule development at early stages.

It was shown that miR444b.1, miRN11, miR166a/165a, and miR952b were differentially expressed between BFs and FMFs at the later stages when morphological differences occurred (Figure 6). In addition, miR444b.1 was differentially expressed between ATNSII and TNSII, and its target, *Gglean003233.1*, encoded a leucine-rich repeat-containing protein and exhibited downregulation as compared with ATNSII and TNSII (Figure 6 and Table 1). Leucine-rich repeat (LRR) proteins are involved in a number of biology processes, including seed and anther development [37,38]. This study focused on the roles of LRR proteins in ovule development. But there have been no reports on the roles of LRR proteins on ovule development so far. In the future, much work is needed to show whether “pg-miR444b.1-*Gglean003233.1*” regulates pomegranate female development. A 22 nt novel miRNA, miRN11, with a relatively abundant read count in TNSII, was not expressed in ATNSII. Its target gene, *Gglean014673.1*, was upregulated in ATNSII as compared with TNSII (Figure 6 and Table 1). *Gglean014673.1* encoded a *KINβ1*, homology of which was reported to be involved in the regulation of sugar metabolism [39], and the latter participated in the determination of plant sexual reproduction [40]. Therefore, “miRN11-*Gglean014673.1*” module can affect pomegranate female sterility by regulating sugar metabolism. 

The miR166a/165a group has been reported to regulate carpel development [41] and its targets, class III homeodomain leucine zipper (HD-ZIP III) transcription factors, play important roles in ovule development. Loss of function of HD-ZIP III genes showed aberrant ovule development in *Arabidopsis* [33]. In this study, pg-miR166a-3p abundantly expressed at FMFs and its expression level gradually increased from ATNSI to ATNSIII. Its targets, *Gglean012177.1* and *Gglean013966.1*, belonging to the HD-ZIP III family, showed the opposite expression patterns. The decrease of *Gglean012177.1* and *Gglean013966.1* expression level could be an important factor affecting ovule development. Moreover, the phenotype of transgenic *A. thaliana* suggested that pg-miR166a-3p influences the development of ovules. Furthermore, pg-miR166a-3p has the capacity to bind to *Gglean012177.1* and *Gglean013966.1*, as confirmed by transient expression. Overall, these results indicated roles of pg-miR166a-3p in modulating pomegranate female sterility. 

### 3.2. MicroRNA Targets Regulating Reproduction Development Involved in Female Sterility

It has been proposed that pomegranate FMFs have been developed with arrested ovules and stigma [8,9], which were involved in floral whorl development. In this study, GO enrichment of differentially expressed miRNA-targets between BFs and FMFs’ pistils revealed that the GO term of “reproduction development” was significantly enriched (*p* < 0.01), suggesting genes involved in pomegranate female sterility. Additionally, *Gglean019708.1* and *Gglean029859.1* were predicted as the targets of pg-miR397a and pg-miR408a, respectively. Similarly, *Gglean012177.1*, *Gglean013966.1*, and *Gglean031286.1* were predicted as pg-miR166a-3p’s targets. These targets were involved in the “reproduction development” process (Appendix A). 

Among these targets, the homologous gene of *Gglean029859.1*, *PLANTACYANIN*, is mainly involved in the anther development and pollination [42], but its role in the pistil development needs to be further confirmed in pomegranate. *Gglean012177.1*, *Gglean013966.1*, and *Gglean031286.1* belong to the HD-ZIPIII gene family, and their expressions can affect the development of the adaxial tissue of lateral organs [43]. *Gglean012177.1* is homologous to the *PHB* gene, which has been involved in the ovule patterning and growth [44] and loss of function of the *PHB* gene displays aberrant integument development [45]. *PHV* (homologous gene of *Gglean013966.1*) and *PHB* overlap in function and expression pattern. Both directly regulate integument development in *A. thaliana* [44]. *REVOLUTA* (*REV*) (homologous gene of *Gglean031286.1*) restricts outer integument [45]. FMFs’ ovule development mainly ceased following the formation of the inner integument primordium [10]. Thus, regulation of these genes affecting integument development can lead to female sterility. *Gglean012177.1* and *Gglean013966.1* showed upregulation in BFs’ pistils as compared with FMFs’ pistils and homologous of them have been reported to have an impact on integument development [45], indicating their role in pomegranate female sterility. 

## 4. Materials and Methods

### 4.1. Sample Collection

In this study, pomegranate pistils were collected from 12-year-old “Tunisia” (also named “Tunisiruanzi”) trees grown in nursery of the Zhengzhou Fruit Research Institute (CAAS) located in Zhengzhou, Henan, China, and managed with conventional methods. Solitary flowers located in perennial branches, which contained a high ratio of BFs (estimated ≥90%), were selected as the sampling location of BFs’ pistils. Lateral flowers located in annual branches, which were observed with a high ratio of FMFs (estimated to be 99%), were selected for FMFs’ pistils (Figure 1a). Previous studies have shown that the key stage for the termination of FMFs’ ovule development was when the bud vertical diameter (BVD) was 5.1 and 13.0 mm [10]. Therefore, BFs and FMFs’ ovary and stigma were cut off from the flowers as pistils according to the BVD, which was measured using a digital caliper. The pistils of FMFs at the pre-stage of FMFs’ ovules aborted (BVD 3.0 to 5.0 mm), the key stage of FMFs’ ovules aborted (BVD 5.1 to 13.0 mm), and the stage post FMFs’ ovules aborted (BVD 13.1 to 25.0 mm) were collected for female sterility related miRNAs mining, and pistils of BFs at the corresponding periods were selected as controls. Each stage with three biology repetitions. 

In order to denote the accessions clearly and easily, the TNSI, TNSII, and TNSIII were used to represent the BFs’ pistils when their BVD was 3.0 to 5.0 mm, 5.1 to 13.0 mm, and 13.1 to 25.0 mm, respectively. Similarly, the designations ATNSI, ATNSII, and ATNSIII were used to represent the FMFs’ pistils when their BVD was 3.0 to 5.0 mm, 5.1 to 13.0 mm, and 13.1 to 25.0 mm, respectively (Figure 1a).

### 4.2. RNA Isolation, Library Construction, and Sequencing

Total RNA was extracted from each sample using CTAB (Solarbio, Beijing, CN) (cetyltrimethyl ammonium bromide) method [46]. The concentration and integrity of total RNA were detected using a NanoDrop 1000 (Thermo Fisher Scientific, Waltham, MA, USA) and 1.5% agarose electrophoresis.

A TruSeq Small RNA Sample Preparation Kit (Illumina, San Diego, CA, USA) was used for the library preparation following the manufacturer’s protocol. The final libraries were sequenced using the Illumina HiSeq 2000 platform (Illumina, San Diego, CA, USA) with 50 bp by IGENECODE (Beijing, China). Finally, more than 20 Mb sequence was generated for each sample.

### 4.3. Bioinformatic Analysis

A total of 18 libraries were obtained. Low quality reads and adaptors from the raw reads were filtered using fastx toolkits [47]. Adaptor sequences, contamination, and low-quality reads, i.e., reads in length ≤15 nt, reads with <3 copies, and junk reads (≥80% A, C, G, or T; ≥3 N; only A, C, or only G/T) were removed from raw data. In view of the length of miRNA (20 to 24 bp), only reads 20 to 24 bp in length and greater than 3 bp in depth were used for further analyses. The correlations among biological replicates were evaluated using “corrplot” package in R [48]. The pearson correlation values for all replications were used to evaluate the consistency of the raw data. Most of them varied from 0.9 to 1.0, except for ATNSI, whose values <0.8 were removed before subsequent analysis. Moreover, small RNAs which were only detected in one of the 18 libraries were also removed before subsequent analysis. The filtered reads for each sample were aligned against the pomegranate ”Tunisia” genome [31] using the Bowtie2 program (Version 2.3.4.1) without mismatch [49]. Only mapped reads were used for further analyses. The conserved known miRNAs were identified with mirdeep2 using miRbase22.1 as the reference [50]. Novel miRNAs prediction was performed using the mireap2 program [50], based on the characteristics of the hairpin structure of the miRNA precursor through exploring the secondary structure. 

### 4.4. MicroRNA Expression Profiles and Comparison between FMFs and BFs

Expression levels of known and novel miRNAs were estimated by read counts. Differential expression analysis of two development stages or two types of floral buds was performed using the DESeq2 R package (1.24.0) [51]. Log2 fold change ≥|0.5| and adjusted *p*-values ≤0.05 were used as the filter criterion of differential expression miRNAs, according to previous reports [52,53,54]. The expression difference level was defined using the following rules: log2fold change ≥ 0.5 or log2fold change ≤−0.5) and adjusted *p*-value ≤ 0.01, extremely significant (**); log2fold change ≥ 0.5 or log2fold change ≤−0.5) and 0.01 < adjusted *p*-value ≤ 0.05, significant (*); otherwise, insignificant.

### 4.5. MicroRNA-target Prediction and Association Analysis with Transcription Analysis Result

Target gene prediction of known and novel miRNA was performed by psRobot (version 1.2) and TargetFinder with default parameters [55,56]. The expressions of the target genes at different development stages were estimated using RNA-seq analyses (RNA-seq analysis was done as in previous work [10]). The RNA-seq and miRNA-seq were performed using the same plant tissue samples [10].

### 4.6. MicroRNA Function Annotation

The known plant miRNAs registered in miRbase (http://www.mirbase.org/) were used for the annotation of the conserved miRNAs. [57]. The GO analysis of target genes was conducted using the AgriGO database (http://bioinfo.cau.edu.cn/agriGO/) [58].

### 4.7. Validation of miRNA Expression in Pomegranate Female Pistils by Stem-Loop qRT-PCR

To accurately evaluate the expression levels of miRNAs, stem-loop qRT-PCR was performed as described by previous studies [29,59] and 5.8S was used as the reference gene [29]. Total RNA was isolated from floral pistils at different development stages, using the approach mentioned above [46]. Quantitative real-time PCR (qRT-PCR) was performed with three biological replicates. All primers used in reverse transcription and qRT-PCR are provided in Appendix A. The relative quantitative expression level was calculated by the 2^−ΔΔ*C*t^ method [59].

### 4.8. Nicotiana Benthamiana Transient Assay

The interactions between miRNAs and their predicted targets were verified by the transient assay in *Nicotiana benthamiana* [60,61,62]. The pg-miR166a-3p precursor was amplified and transferred into the PBI121-GUS vector with an empty vector (EV) as the control. PBI121-GUS EV was used to eliminate the influence of the sequence in the vector to the target site. The native target sites (5′- TTGGGATGAAGCCTGGTCCGG-3′ in *Gglean012177.1* and 5′-CTCTGGGATGAAGCCTGGTCCGGC-3′ in *Gglean013966.1*) modified target sites (5′-CTGAGCGAGGATAGCAGACGG-3′) that could not be cleaved by pg-miR166a-3p, and perfect complementary sites (5′- TCGGGGAATGAAGCCTGGTCCGAG-3′) were inserted into the pMS4 vector (kindly provided by Dr Fengquan Tan, Huazhong Agricultural University, Wuhan, China) carrying green fluorescent protein (GFP) (Figure 4). Pg-miR166a-3p and its target sites were transformed into *Agrobacterium* strain GV3101 and co-infiltrated into the *Nicotiana benthamiana* leaves by a ratio of 4:1 (pre-pg-miR166a-3p-PBI121: target sites-pMS4). The fluorescence was captured under a hand-held UV light (Shanghai Guanghao Analytical instrument co. LTD, Shanghai, China). Pre-pg-miR166a-3p-PBI121 was transformed to Columbia *A. thaliana* as reported method [63]. Wild type and transgenic *A. thaliana* were sampled from inner to outer rind when the first inflorescence flowers. The paraffin sections were stained with hematoxylin. The morphology of the ovules of 35S-pre-pg-miR166a-3p were captured by Olympus DP71 (Olympus, Tokyo, Japan).

## 5. Conclusions

The current study provides a genome-wide comparison of miRNAs and their targets among different stages of FMFs’ pistils, among different stages of BFs’ pistils, and between FMFs’ and BFs’ pistils at the same stage. Combined analyses of the small RNA and mRNA revealed that pg-miR858b and pg-miRN01 affect pomegranate pistil development at the early stage (before inner integument primordium formation); pg-miR444b.1, pg-miRN11, pg-miR166/165a-3p, and pg-miR952b influence pomegranate integument and embryo sac development (stages after inner integument primordium formation). In addition, pg-miR166a-3p and its predicted target genes, *Gglean012177.1* and *Gglean013966.1*, showed reciprocal expression patterns. A. thaliana transformed with 35S-pre-pg-miR166a-3p verified the role of pg-miR166a-3p in ovule development, and the transient assay in *Nicotiana benthamiana* confirmed the interaction between pg-miR166a-3p and *Gglean012177.1* and *Gglean013966.1*. In conclusion, this study provides new insight into the miRNA-mediated network models that could regulate pomegranate female sterility.

## Figures and Tables

**Figure 1 ijms-21-00558-f001:**
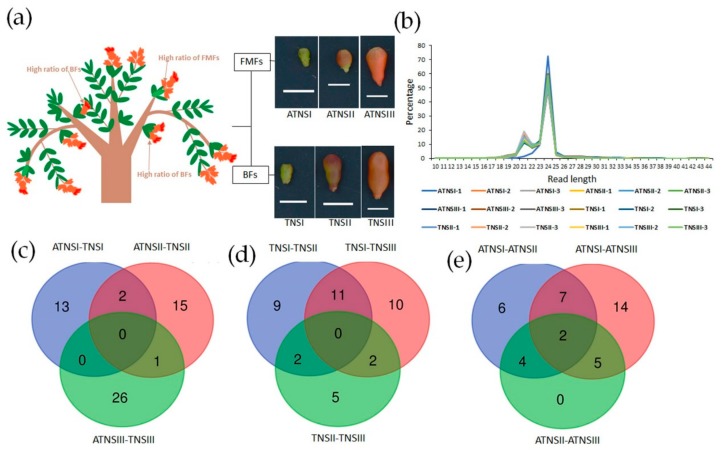
Schematic representation of sampling location and overview of the high-throughput sequencing data: (**a**) The locations with high ratio of FMFs and BFs and the external morphology of sampling stages, (**b**) length distribution of small RNA reads in the sequencing samples, (**c**–**e**) Venn diagram showing unique and shared differentially expressed miRNAs, (**c**) differentially expressed miRNAs between FMFs and BFs, (**d**) differentially expressed miRNAs among FMFs development stages, and (**e**) differentially expressed miRNAs among BFs development stages. FMFs, functional male flowers and BFs, bisexual flowers, TNSI, TNSII, and TNSIII represent the BFs’ pistils when their vertical diameters were 3.0 to 5.0 mm, 5.1 to 13.0 mm, and 13.1 to 25.0 mm, respectively. Similarly, the designations ATNSI, ATNSII, and ATNSIII were used to represent the FMFs’ pistils when their vertical diameters were 3.0 to 5.0 mm, 5.1 to 13.0 mm, and 13.1 to 25.0 mm, respectively. Scale bars = 1.0 cm.

**Figure 2 ijms-21-00558-f002:**
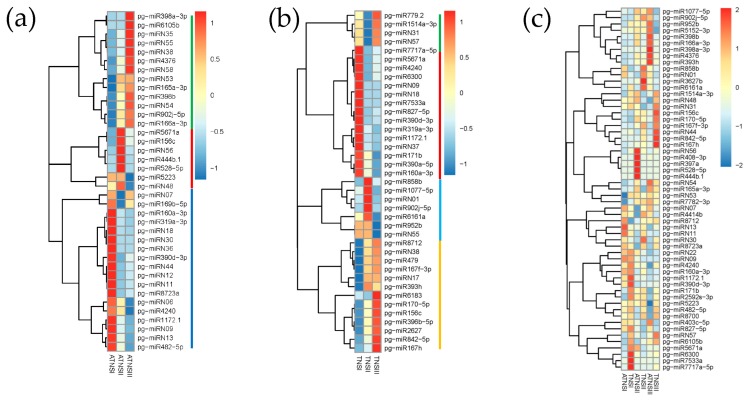
Sequencing frequency profiles of the known miRNAs and novel miRNAs: (**a**) Different expressed miRNAs among different stages of FMFs, the green line represents a group of miRNAs highly expressed in ATNSIII, the red line represents a group of miRNAs highly expressed in ATNSII, and the blue line represents a group of miRNAs highly expressed in ATNSI; (**b**) different expressed miRNAs among different stages of BFs, the green and yellow lines represent a group of miRNAs highly expressed in TNSIII, the red line represents a group of miRNAs highly expressed in TNSI, and the blue line represents a group of miRNAs highly expressed in TNSII; (**c**) different expressed miRNAs between FMFs and BFs. ATNSI-ATNSIII, FMFs’ pistils when their bud vertical diameters (BVDs) were 3.0 to 5.0 mm, 5.1 to 13.0 mm, and 13.1 to 25.0 mm, respectively, and TNSI-TNSIII, BFs’ pistils when their BVDs were 3.0 to 5.0 mm, 5.1 to 13.0 mm, and 13.1 to 25.0 mm, respectively.

**Figure 3 ijms-21-00558-f003:**
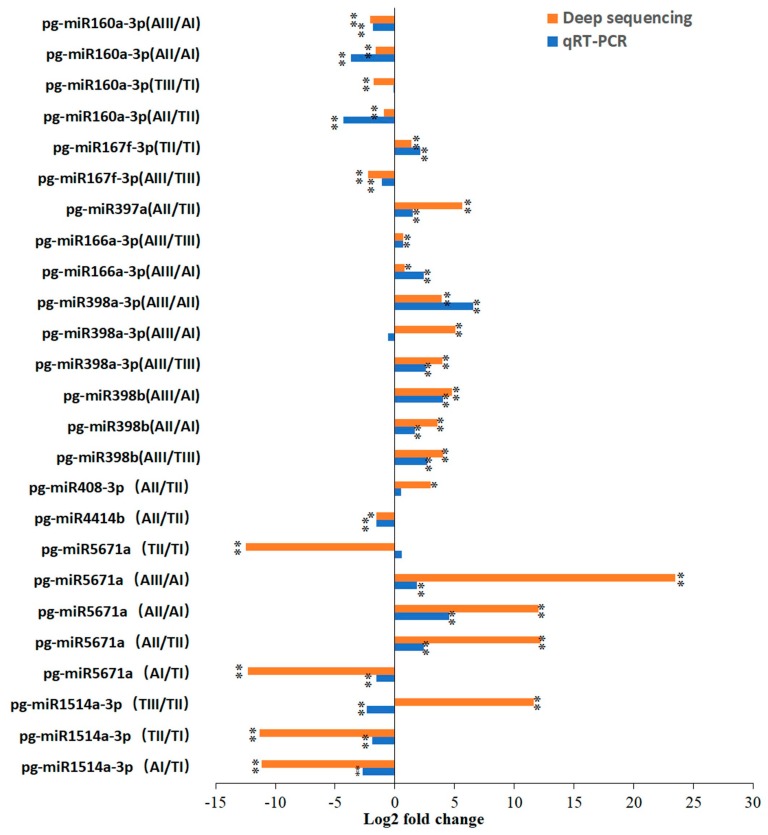
The qRT-PCR validation of differential expression patterns of miRNAs between different stages. **p* < 0.05 and ***p* < 0.01 (Student’s *t*-test).

**Figure 4 ijms-21-00558-f004:**
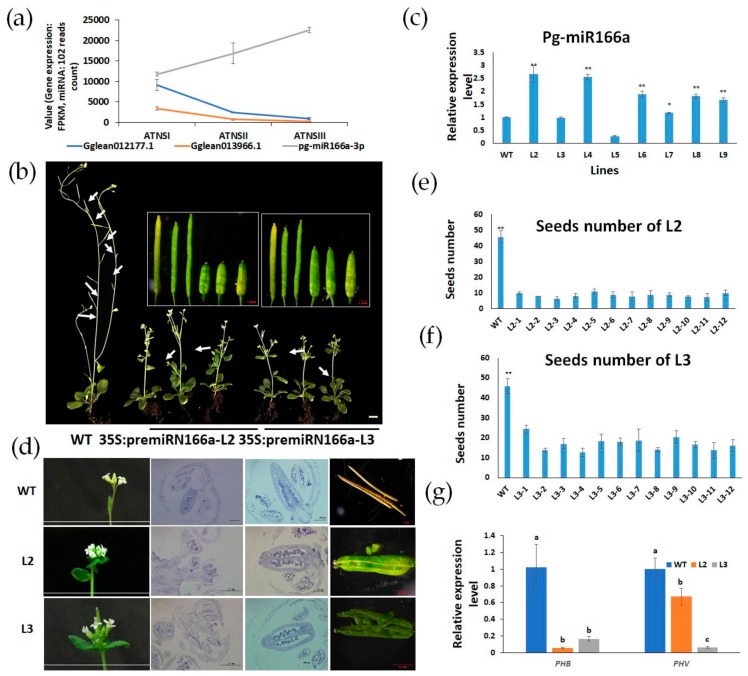
The phenotypes of 35S-pre-pg-miR166a-3p: (**a**) Expression level of pg-miR166a-3p, *Gglean012177.1*, and *Gglean013966.1* during the FMFs development; (**b**) the external morphology of 35S-pre-pg-miR166a-3p and wild type *A. thaliana*, the white arrow shows the location of seeds; (**c**) relative expression level of pg-miR166a-3p in different transgenic lines; (**d**) the internal morphology of 35S-pre-pg-miR166a-3p and wild type Arabidopsis; (**e**) seeds number of 35S-pre-pg-miR166a-3p line 2; (**f**) seeds number of 35S-pre-pg-miR166a-3p line 3; and (**g**) the expression level of homology genes of Gglean012177.1 (*PHB*) and *Gglean013966.1* (*PHV*) in wild type *A. thaliana*, L2, and L3.

**Figure 5 ijms-21-00558-f005:**
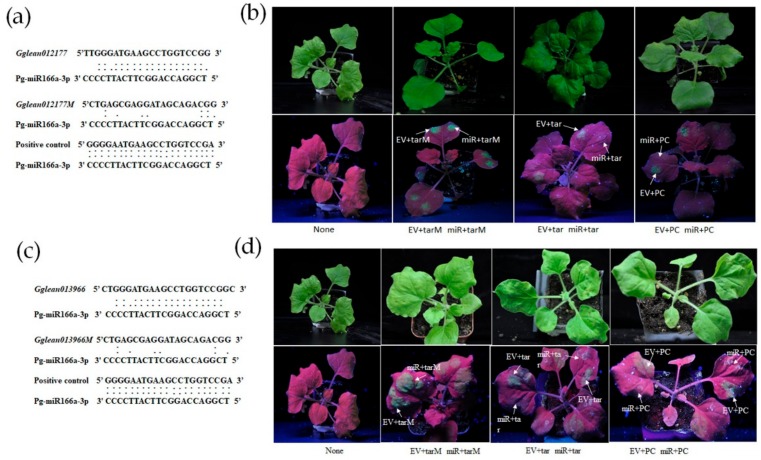
Pg-miR166a-3p targeted *Gglean012177.1* and *Gglean013966.1* was verified in vivo: (**a**) The alignment of the sequence of pg-miR166a-3p with the target site in *Gglean012177*, the negative control and the positive control; (**b**) verification of the interaction between pg-miR166a-3p and *Gglean012177.1*; (**c**) the alignment of the sequence of pg-miR166a-3p with the target site in *Gglean013966*, the negative control and the positive control; and (**d**) verification of the interaction between pg-miR166a-3p and *Gglean013966.1*. ”:” in the picture represent the complementary site; None, tobacco without anything injection; EV, PBI121 empty vector; tarM, the negative control (*Gglean012177M/Gglean013966M*); tar, the target site of pg-miR166a-3p in *Gglean012177.1* and *Gglean013966.1;* miR, pre-pg-miR166a-3p-PBI121; and PC, the positive control.

**Figure 6 ijms-21-00558-f006:**
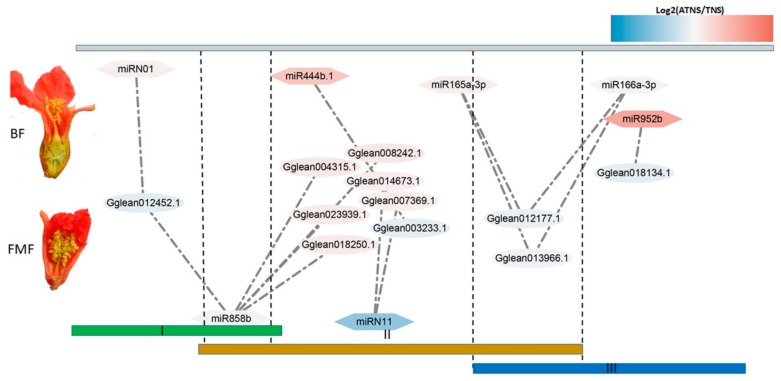
Pomegranate miRNA-mediated interaction network during BFs and FMFs pistil development stages. I, ATNSI and TNSI; II, ATNSII and TNSII; and III, ATNSIII and TNSIII.

**Table 1 ijms-21-00558-t001:** Predicted mRNA targets of differentially expressed miRNAs in ATNSI_TNSI, ATNSII_TNSII, and ATNSIII_TNSIII. Note: UP represents upregulation and DOWN represents downregulation.

miRNA ID	*p*-Value	Log2 (ATNS/TNS)	Regulation	Target Gene ID	*p*-Value	Log2 (ATNS/TNS)	Regulation
ATNSI/TNSI							
miR858b	0.0005004	0.921592594	UP	*Gglean012452.1*	0.0052191	−2.680132231	DOWN
miRN01	0.0000002	1.432745067	UP	*Gglean012452.1*	0.0052191	−2.680132231	DOWN
ATNSII/TNSII							
miRN11	0.0093121	−13.15241036	DOWN	*Gglean007369.1*	0.0136615	1.678181538	UP
miRN11	0.0093121	−13.15241036	DOWN	*Gglean014673.1*	0.0043412	1.779818526	UP
miR165a-3p	0.0150903	1.861219672	UP	*Gglean013966.1*	0.0000209	−1.188541887	DOWN
miR165a-3p	0.0150903	1.861219672	UP	*Gglean012177.1*	0.0001133	−1.099146097	DOWN
miR858b	0.0005004	−0.863522677	DOWN	*Gglean008242.1*	0.000037	2.754373731	UP
miR858b	0.0005004	−0.863522677	DOWN	*Gglean018250.1*	0.000000251	2.372360824	UP
miR858b	0.0005004	−0.863522677	DOWN	*Gglean023939.1*	0.0020669	2.157568169	UP
miR858b	0.0005004	−0.863522677	DOWN	*Gglean004315.1*	0.00000026	2.769310866	UP
miR444b.1	0.0482444	8.687029035	UP	*Gglean003233.1*	0.0001883	−2.440374727	DOWN
ATNSIII/TNSIII							
miR952b	0.000000000000000000000509	14.53397128	UP	*Gglean018134.1*	0.0004515	−2.379136194	DOWN
miR166a-3p	0.00000000000000000491	0.718106582	UP	*Gglean012177.1*	0.0000015	−1.614832343	DOWN
miR166a-3p	0.00000000000000000491	0.718106582	UP	*Gglean013966.1*	0.0077827	−1.215622716	DOWN

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
