# Peer review of "Small RNA and mRNA Sequencing Reveal the Roles of microRNAs Involved in Pomegranate Female Sterility"

_ijms, 2020, doi:10.3390/ijms21020558_

Round 1

Reviewer 1 Report

The current study “Small RNA and mRNA sequencing reveal roles of miRNAs involved in pomegranate female sterility“ is on a topic of relevance and general interest to the readers of the journal. I found the paper to be overall well written and felt confident that the authors performed careful and thorough field and spectral processing. I have several significant concerns about the presentation and general results that should be addressed prior to publication (See my comments on the MS). For example, Figure 4 and Figure 6 have some overlapping information, as I suggested these 2 figures could be merged into one figure.

The sequences of the figures in the text somehow make it difficult to follow, the author is advised to rearrange the figure to be presented in sequence in the text.

See all of my other comment on the MS directly!

Author Response

We appreciated for your comments and suggestions on our manuscript, and we have done the main changes in light of your detailed and helpful comments as follows:

Title: miRNAs have been changed to microRNAs in the title. The Grammatical & spelling mistakes and the use of personal pronounces were revised throughout the manuscript in the revised manuscript. Key words: key words were changed to “High-throughput sequencing; functional male flowers; bisexual flowers; reproduction process; andromonoecy; female sterility; ovule”. Figures: the sequence of the figures presented in the text were revised in the revised manuscript. Table S2: the novel miRNAs were highlighted in Table S2. Figure 3: significance marks were added to Figure 3. Table 1: all abbreviation in the table were defined in the note. Figure 4 and Figure 6: figure 4 and figure 6 were merged in one figure in the revised manuscript. 4.2 section: the name and the source of the recovery kit was added. 4.3 section: “corrplot” package was used for coefficient analysis in R, and we have added the name of the package in the 4.3 section. Abbreviations: all the abbreviations were deleted in the revised manuscript.

Reviewer 2 Report

This work provides a genome-wide approach of miRNAs during sterile and fertile pomegranate flower development. Many miRNA have been detected differentially expressed and 58 novel have been described. By comparing this data with mRNA previously published data, the authors describe the most interesting miRNA-target modules that could control sterility during pistil development.

I have some comments to the authors:

1) Can the authors clarify if in the moment of the dissection of the pistils they can distinguish if the flower will be FMF or BF? If they rely in the high ratio of one or the other one depending on where the flowers are placed, can they estimate this ratio?

2) When the authors identify the 161 miRNAs (section 2.2) from the pistils, the major group is the 21nt (92 out of 161), meanwhile the group of 24nt is only 25 of 166. This is not in agreement with the Fig1b, where most of the miRNA are 24nt. Can the authors comment on that? Is this related with their expression? Still in the Fig1b, when observing the 21nt class, there are samples with values around 10% and others around 20%. Is this not biologically important? Finally, in this figure the ATNSI-1 sample is missing.

3) The Fig3 does not have any standard deviation/error (although either miRNA-seq and qPCR were done with biological triplicates). The significant comparisons based on the t-test applied are not marked (or all are not significant?). The criteria for the selection of the type of comparison in each miRNA should be explained (i.e. why there is only the AII/TII comparison in the miR397a, but in other miRNA there are other ones?).

4) For the miRNA-mRNA identification, of the 144 genes targeted (Table S8), only few were negatively correlated (Table 1). How can this be explained?

5) In the experiment of overexpressing pg-miR166a-3p in A. thaliana the authors selected two lines that present very similar phenotype although the relative expression levels differ: one of the lines is truly overexpressed but the other has the same levels than WT. The authors do not explain how this is possible. How can be explained the similar phenotype if the expression of the miRNA is different? In the same experiment, in the interpretation of the result, can the authors develop which are the putative targets of miR166a-3p in Arabidopsis that could explain the phenotype?

6) The section 3.1 of the discussion is almost exclusively a description of results. That section can fit better in the Results.

7) How miRNA triplicates have been treated? In the Bioinformatic analysis section of the Materials and Methods is not explained. Neither in the table S2. Is the value of each miRNA in this table an average, a sum?

Minor comments:

Line 62. ARF6 and ARF8 genes do not have the full name, as done later with others like PHABULOSA (PHB).

Line 68. Missing the full name of SPL/SBP.

Line 73. Missing the full name of INO, ANT.

Line 83, 86. The samples or libraries (referred like this in methods and Table S1) are referred here as “accessions”, instead of libraries/samples.

Line 93. There are two dots finishing the sentence.

Line 189. It is not clear what “respectively” refers to.

Line 197. In the section 2.6 refers to Fig5 when should be Fig4. The order of the figures should be shifted as Fig5 cannot be mention before the Fig4.

Line 221. It is referred that two genes “were enriched in reproductive structure process”, but this cannot be possible. They can belong to this GO, but only lists of genes can be enriched in GO terms. The same in line 321.

Line 295. The study focused on the roles of LRR proteins is not referred.

Line 359. “integrality” should be “integrity”.

Line 362. The recovery kit used is not referred. Neither the adaptors used (or the kit used for the library preparation).

Line 373. The coefficient package in R is not referred.

Line 376. “was” should be “were”.

Line 387. The definition of (*) is unclear.

Line 391. How are the RNA-seq analysis done? Maybe as in the previous work [9]?

Line 407. The role of the empty vector here is not clear.

Line 428, 429, 431. Genes and species should be in italics.

Line 434. The supplementary materials are not in the Supplementary data provided.

Fig. 2. In the Fig2a and b, the colored line between the miRNA names and the color reference is not described what is for. The units of the scale are not explained neither. The samples with no values can be marked in gray or white instead of colored in blue, otherwise can be interpreted as a negative value.

Fig4. Panel (a) and (b) has too small font size. It is not explained in the legend what tarM, tar, PC… means. The same apply to Fig6.  One suggestion is to unify Fig4 and 6 as they are totally related, and they can share information. Figure legends, especially in Fig6 are lacking the description of the panels and many details are missing.

Fig7. BF and FMF flowers are not identified with the pictures.

Tables S3-S5, miRNAs are capitalized (“Pg-miR…”). The definition of “padj” is missing.

Table S2. “Sterility” should be “sterile”.

Table S6. What “NA” stands for?

Table S7. What means DEM should be defined (not used in the text, only in the table title). What is the table in the “sheet 2”?

Author Response

Dear Reviewer:

We appreciated for your detailed and helpful comments for our manuscript, and now we have extensively revised our manuscript, the descriptions on the modification are as follows:

Point 1: Can the authors clarify if in the moment of the dissection of the pistils they can distinguish if the flower will be FMF or BF? If they rely in the high ratio of one or the other one depending on where the flowers are placed, can they estimate this ratio?

Response 1: Previous investigations showed that there was no obvious difference between the ATNSI and TNSI in appearance, but the FMF or BF can be distinguished by the shape and the height of the stigma at the second and third stage (ATNSII, ATNSIII, TNSII, TNSIII). Based on the previous investigations, the ratio of BFs of solitary flowers located in perennial branches could be more than 90%. The ratio of FMFs of lateral flowers located in annual branches could reach to 99%. So, solitary flowers located in perennial branches were selected as the sampling location of BFs’ pistils, lateral flowers located in annual branches were selected for FMFs’ pistils.

Point 2: When the authors identify the 161 miRNAs (section 2.2) from the pistils, the major group is the 21nt (92 out of 161), meanwhile the group of 24nt is only 25 of 166. This is not in agreement with the Fig1b, where most of the miRNA are 24nt. Can the authors comment on that? Is this related with their expression? Still in the Fig1b, when observing the 21nt class, there are samples with values around 10% and others around 20%. Is this not biologically important? Finally, in this figure the ATNSI-1 sample is missing.

Response 2: To obtain the reliable results, correlations among the biological replicates were evaluated, and biological replicates with Pearson correlation coefficient smaller than 0.80 were removed. Finally, the Pearson correlation coefficient of ATNSI-1 with ATNSI-2 and ATNS1-3 was < 0.8, therefore only ATNSI-2 and ATNS1-3 was used for further analyses. Moreover, small RNAs which were detected only in 1 of the 18 libraries were also removed before subsequent analysis. These are reasons why the major group showed in section 2.2 was not agreement with Fig.1b.

     Fig.1b showed the length distribution of all detected small RNAs, and we have added the ATNSI-1 sample to the Fig 1b in the revised manuscript. Moreover, we have supplemented to the description about the filtering of specific data in 4.3.

    The reason why the values around 10% and others around 20% was the proportion of sRNAs with different length varied among different stages.  

Point 3: The Fig3 does not have any standard deviation/error (although either miRNA-seq and qPCR were done with biological triplicates). The significant comparisons based on the t-test applied are not marked (or all are not significant?). The criteria for the selection of the type of comparison in each miRNA should be explained (i.e. why there is only the AII/TII comparison in the miR397a, but in other miRNA there are other ones?).

Response 3: We are sorry for the missing of the significance marks, and we have added them to the Fig. 3 in the revised manuscript. The expression of 10 miRNAs randomly selected miRNAs were validated using stem-loop qRT-PCR, Fig.3 only showed the comparison with significant differences in sequencing result. The significant of miRNAs in these comparison by stem-loop qRT-PCR were compared with the sequencing result.

Point 4: For the miRNA-mRNA identification, of the 144 genes targeted (Table S8), only few were negatively correlated (Table 1). How can this be explained?

Response 4: miRNAs participate in a variety of biological processes, and many studies have shown that the same miRNA can target multiple genes, the same gene can be targeted by multiple miRNAs. On the other hand, except for miRNAs, genes may be regulated by other regulation genes. We think these may be the mainly reasons leading to only few were negatively correlated. Thank you for your comments, and we have added the discussion to 3.3 section.

Point 5: In the experiment of overexpressing pg-miR166a-3p in A. thaliana the authors selected two lines that present very similar phenotype although the relative expression levels differ: one of the lines is truly overexpressed but the other has the same levels than WT. The authors do not explain how this is possible. How can be explained the similar phenotype if the expression of the miRNA is different? In the same experiment, in the interpretation of the result, can the authors develop which are the putative targets of miR166a-3p in Arabidopsis that could explain the phenotype?

Response 5: Except for L5, all other transgenic lines with 35S::pg-miR166a-3p exhibited declining in plant biomass, clip size, and seeds number. The expression level of pg-miR166a-3p in L2 was significantly higher than L3. We have observed 12 individual plants of L2 and L3 respectively, the result showed that the number of seeds of L2 was significantly less than L3 (Table A), and the plant height and fruit clamping length were significantly smaller than L3 (Figure A). The morphological difference may due to the different expression level between L2 and L3. Table A and Figure A were attached to the attachment.

      On the other hand, the expression of homology genes of Gglean012177.1 (PHB) and Gglean013966.1 (PHV) in A. thaliana were detected by qRT-PCR, the results showed that the expression of both PHB and PHV in L2/L3 was significantly lower than in WT, which indicated that both PHB and PHV were the putative targets of pg-miR166a-3p (Figure B). Thank you for your suggestion, and the result of the expression of the putative targets have been added to 2.6 section in revised manuscript. Figure B were attached to the attachment.

Point 6: The section 3.1 of the discussion is almost exclusively a description of results. That section can fit better in the Results.

Response 6: Thank you for your suggestion, and we have changed 3.1 section to the result section 2.7.

Point 7: How miRNA triplicates have been treated? In the Bioinformatic analysis section of the Materials and Methods is not explained. Neither in the table S2. Is the value of each miRNA in this table an average, a sum?

Response 7: Thank you for your comments, correlations among biological replicates were evaluated, and biological replicates with Pearson correlation coefficient smaller than 0.80 were removed. In addition, small RNAs which were only detected in 1 of the 18 libraries were also remove prior to subsequent analysis. The description has been added to the Bioinformatic analysis section in the revised manuscript. Table S2 showed the average value of each miRNA, and we have appended the annotation in the table note.

Point 8: Line 62. ARF6 and ARF8 genes do not have the full name, as done later with others like PHABULOSA (PHB).

Response 8: Thank you for your comment, and we have added Auxin Response Factors 6 (ARF6) and Auxin Response Factors 8 (ARF8) to the revised manuscript.

Point 9: Line 68. Missing the full name of SPL/SBP.

Response 9: Thank you for your comment, and we have added SQUAMOSA PROMOTER BINDING PROTEIN (SPL/SBP) to the revised manuscript.

Point 10: Line 73. Missing the full name of INO, ANT.

Response 10: Thank you for your comment, and we have added INNER NO OUTER (INO), AINTEGUMENTA (ANT) to the revised manuscript.

Point 11: Line 83, 86. The samples or libraries (referred like this in methods and Table S1) are referred here as “accessions”, instead of libraries/samples.

Response 11: Thank you for your comment, and we have changed ‘accessions’ to ‘libraries/samples’ in the revised manuscript.

Point 12: Line 93. There are two dots finishing the sentence.

Response 12: Thank you for your comment, and we have deleted one of the dots in the revised manuscript.

Point 13: Line 189. It is not clear what “respectively” refers to.

Response 13: Thank you for your comment, and we have changed the sentence to “Pg-miR165a-3p negatively regulated the expression of Gglean013966.1, Gglean012177.1 and pg-miR444b.1 repressed the expression of Gglean003233.1 during the key stage of pomegranate female sterility (ATNSII_TNSII) (Table 1).”

Point 14: Line 197. In the section 2.6 refers to Fig5 when should be Fig4. The order of the figures should be shifted as Fig5 cannot be mention before the Fig4.

Response 14: Thank you for your comment, and we have changed the order of the figures in the revised manuscript.

Point 15: Line 221. It is referred that two genes “were enriched in reproductive structure process”, but this cannot be possible. They can belong to this GO, but only lists of genes can be enriched in GO terms. The same in line 321.

Response 15: Thank you for your comment, and we have changed ‘enriched’ to ‘involved’ in the revised manuscript.

Point 16: Line 295. The study focused on the roles of LRR proteins is not referred.

Response 16: Thank you for your comment. Leucine-rich repeat (LRR) proteins are involved in a number of biology processes, including seed and anther development, but there were no reports on the roles of LRR proteins on ovule development so far. In future, much work will be needed to show whether ‘pg-miR444b.1- Gglean003233.1’ regulate pomegranate female sterility.

Point 17: Line 359. “integrality” should be “integrity”.

Response 17: Thank you for your comment, and we have changed “integrality” to “integrity” in the revised manuscript.

Point 18: Line 362. The recovery kit used is not referred. Neither the adaptors used (or the kit used for the library preparation).

Response 18: Thank you for your comment, the library was prepared by IGENECODE (Beijing, China), TruSeq Small RNA Sample Preparation Kit (Illumina, San Diego, CA) was used for the library preparation.  

Point 19: Line 373. The coefficient package in R is not referred.

Response 19: Thank you for your comment, “corrplot” package were used for coefficient analysis, and we have added it to 4.3 section.

Point 20: Line 376. “was” should be “were”.

Response 20: Thank you for your comment, and we have changed “was” to “were”.

Point 21: Line 387. The definition of (*) is unclear.

Response 21: Thank you for your comment, and we have revised it.

Point 22: Line 391. How are the RNA-seq analysis done? Maybe as in the previous work [9]?

Response 22: RNA-seq analysis was done as previous work [9], we have noted this in 4.5 section.

Point 23: Line 407. The role of the empty vector here is not clear.

Response 23: The role of the empty vector was to eliminate the influence of the sequence in the vector to the target site. We have added the explain to 4.8 section.

Point 24: Line 428, 429, 431. Genes and species should be in italics.

Response 24: Thank you for your comment, and we have revised them.

Point 25: Line 434. The supplementary materials are not in the Supplementary data provided.

Response 25: Thank you for your comment, and the supplementary figures and tables are available online.

Point 26: Fig. 2. In the Fig2a and b, the colored line between the miRNA names and the color reference is not described what is for. The units of the scale are not explained neither. The samples with no values can be marked in gray or white instead of colored in blue, otherwise can be interpreted as a negative value.

Response 26: Thank you for your comments, the colored lines between the miRNA names and the color reference represent the clusters of miRNAs with similar expression patterns. The heat maps were plotted using the normalized data. Therefore, color reference was the Row z-scale, and the red indicated the high expression level, while blue indicated the low expression level.

Point 27: Fig4. Panel (a) and (b) has too small font size. It is not explained in the legend what tarM, tar, PC… means. The same apply to Fig6. One suggestion is to unify Fig4 and 6 as they are totally related, and they can share information. Figure legends, especially in Fig6 are lacking the description of the panels and many details are missing.

Response 27: Thank you for your suggestions, we have combined Fig.4 and Fig.6 together. what tarM, tar, PC, EV, miR means were explained in the figure legend in the revised manuscript.

Point 28: Fig7. BF and FMF flowers are not identified with the pictures.

Response 28: Thank you for your comment, BF and FMF were remarked in Fig.7.

Point 29: Tables S3-S5, miRNAs are capitalized (“Pg-miR…”). The definition of “padj” is missing.

Response 29: Thank you for your comment, we have changed all the name of miRNAs to “Pg-miR…”, padj represented the adjusted P-value, we have noted it in the revised manuscript.

Point 30: Table S2. “Sterility” should be “sterile”.

Response 30: Thank you for your comment. We have modified it in revised manuscript.

Point 31: Table S6. What “NA” stands for?

Response 31: Thank you for your comment. NA represent values were filtered:-0.5<log2fold change<0.5  and adjusted P-value>0.05

Point 32: Table S7. What means DEM should be defined (not used in the text, only in the table title). What is the table in the “sheet 2”?

Response 32: DEM represented differentially expressed miRNA, we have added the full name to Table S7. We are sorry for the dropping down of “sheet 2”, we have deleted it in the revised manuscript.
